# Spatiotemporal Dynamics and Drivers of Vegetation Carbon Sinks in Zhejiang Province: A Case Study in Rapidly Urbanizing Subtropical Ecosystems

**DOI:** 10.3390/plants14071151

**Published:** 2025-04-07

**Authors:** Juntao Xu, Nguyễn Thị Hằng, Mengqi Ran, Junqia Kong

**Affiliations:** College of Landscape Architecture, Zhejiang A&F University, Hangzhou 311300, China; xjt@stu.zafu.edu.cn (J.X.); hangnt4778@gmail.com (N.T.H.); 15290403901@163.com (M.R.)

**Keywords:** Net Ecosystem Productivity (NEP), vegetation carbon sink, driving factors, spatiotemporal dynamics, Zhejiang Province

## Abstract

As a national ecological civilization pilot, Zhejiang’s growing vegetation carbon sink capacity is important for both regional ecological security and China’s carbon neutrality goals, but current studies lack a comprehensive assessment of multi-factor interactions. This study employed an improved Carnegie–Ames–Stanford Approach (CASA) and soil respiration empirical equation to estimate Net Ecosystem Productivity (NEP) in Zhejiang Province, and trend analysis, partial correlation analysis, and the GeoDetector model based on optimal parameters (OPGD) were utilized to investigate the spatiotemporal variations and driving factors of vegetation NEP. The results showed that the multi-year average NEP and carbon sink capacity in Zhejiang Province were 387.67 g C m^−2^ a^−1^ and 38.84 Tg C a^−1^, exhibiting an increasing trend at an average rate of 2.15 g C m^−2^ a^−1^ and 0.23 Tg C a^−1^, respectively. Spatially, NEP was higher in the western and southern mountainous regions and lower in the eastern coastal and northern plains. NEP in Zhejiang Province was driven by both natural and anthropogenic factors, with NDVI (q = 0.502) and elevation (q = 0.373) being the primary natural drivers, and nighttime light intensity (q = 0.327) and impervious surface dynamics (q = 0.295) being the main anthropogenic drivers. Moreover, the interactions among these factors all exhibited synergistic enhancement effects. Overall, Zhejiang Province functioned predominantly as a carbon sink, with its sequestration capacity gradually strengthening over time. The combined effects of natural and anthropogenic factors drove the spatiotemporal heterogeneity of vegetation NEP. These findings highlight the importance of coordinated ecosystem management strategies that consider both natural and anthropogenic-induced impacts to enhance the achievement of regional carbon sink goals.

## 1. Introduction

Terrestrial ecosystem carbon cycling, governed by the equilibrium between photosynthetic carbon sequestration and soil respiratory carbon release, plays a pivotal role in regulating atmospheric CO_2_ concentrations and global climate dynamics [1]. As a fundamental component of the global carbon budget, this process determines the net balance between carbon sinks and sources. However, escalating anthropogenic pressures (including land use conversion, deforestation, and urban expansion) have increasingly destabilizing this delicate equilibrium [2]. Consequently, comprehensive quantification of terrestrial carbon flux dynamics becomes imperative for advancing carbon cycle science, refining climate change projections, and achieving carbon neutrality goals.

NEP, defined as the residual carbon flux derived from Net Primary Productivity (NPP) minus soil heterotrophic respiration (Rh), serves as a critical metric for assessing ecosystem carbon sink–source transitions [3]. Spatiotemporal analysis of NEP enables the characterization of key ecological processes, including carbon allocation patterns, storage capacity, and energy transfer efficiency across diverse biomes [4]. Thus, mapping regional NEP dynamics holds significant implications for guiding ecosystem restoration, optimizing land management strategies, and advancing sustainable development objectives.

Recent research on the spatiotemporal distribution of regional NEP has achieved significant advancements in both methodological and applied dimensions [5]. The integration of multi-source remote-sensing data (e.g., Moderate-Resolution Imaging Spectroradiometer Normalized Difference Vegetation Index (MODIS NDVI), Landsat Thematic Mapper/Operational Land Imager (TM/OLI)) with process-based models (e.g., CASA, Boreal Ecosystem Productivity Simulator (BEPS)) has enabled high-resolution mapping of carbon sink dynamics through synergistic approaches combining light use efficiency modeling for NPP estimation and empirical parameterization of Rh [6]. These advancements have particularly facilitated NEP quantification in heterogeneous landscapes, yielding critical datasets for regional carbon budget assessments [7]. However, the mechanisms underlying spatiotemporal differentiation of NEP in complex ecosystems, such as rapidly urbanizing regions, remain underexplored [8]. Factors such as terrain fragmentation, vegetation type diversity, and anthropogenic disturbances (e.g., arable land expansion, impervious surface proliferation) pose challenges in localizing model parameterization [9]. For example, the applicability of the CASA model in vegetation–urban transition zones (e.g., Zhejiang Province, China) is often constrained by the need for light use efficiency calibration in heterogeneous habitats [10]. Moreover, the model lacks extensive and refined application in such contexts [11]. Therefore, adopting NEP estimation models suitable for complex ecosystems (e.g., rapidly urbanizing regions) and conducting quantitative analyses of spatiotemporal evolution in vegetation carbon sinks will not only address theoretical gaps in carbon cycle research for these areas but also provide scientific tools for ecological restoration and land management in highly heterogeneous habitats.

The spatiotemporal differentiation of NEP is jointly driven by natural factors and human activities, with significant regional heterogeneity and multi-scale characteristics in their mechanisms [12]. Among natural factors, climatic conditions (e.g., temperature, precipitation, and solar radiation) directly influence the spatial patterns of NEP by regulating the rates of vegetation photosynthesis and soil respiration [13]. For instance, drought stress may weaken the carbon sink capacity by reducing NPP [14]. Topographic factors (e.g., elevation and slope) indirectly affect the vertical differentiation of NEP by altering hydrothermal redistribution and vegetation types [15]. Among anthropogenic factors, land use changes (e.g., deforestation and arable land expansion) directly change the potential for vegetation carbon sequestration by altering land cover types [16]. Concurrently, urbanization-driven impervious surface expansion and increased nighttime light intensity may suppress vegetation growth through heat island effects and habitat fragmentation [17]. Recent applications of interaction detection methods (e.g., GeoDetector) and multi-model coupling (e.g., Structural Equation Modeling, SEM) have shown that the combined effects of natural and human factors may dominate the dynamic changes in NEP [18]. However, most existing studies have focused on the independent contributions of single driving factors, with insufficient analysis of the nonlinear interactions among multiple factors [19]. This limits a comprehensive understanding of the integrated impacts of natural and human drivers, particularly in eco–economic transition zones (e.g., rapidly urbanizing regions), where the mechanisms of compound impacts of human activities and natural stresses still need to be further explored [20]. These methodological and theoretical gaps limit the formulation of targeted carbon management policies in rapidly urbanizing regions.

Zhejiang Province, situated in China’s southeastern coastal region, exhibits a subtropical monsoon climate that sustains diverse vegetation types and complex ecosystem structures [21]. This economically vibrant region has undergone substantial land use transformations during rapid urbanization and industrialization, resulting in intensified anthropogenic pressures on vegetative ecosystems [22]. Notably, as a national pilot zone for ecological civilization initiatives, Zhejiang’s evolving vegetation carbon sequestration capacity carries dual significance—both influencing regional ecological security and contributing to China’s broader carbon neutrality goals [23]. Current research gaps persist regarding NEP in Zhejiang, particularly in comprehensive assessments of multi-factorial interactions, as existing studies predominantly employ single-factor analytical approaches [24]. Therefore, this study focuses on the rapidly urbanizing subtropical ecosystem in Zhejiang Province and aims to achieve several key objectives: (1) to estimate Net Ecosystem Productivity (NEP) in heterogeneous ecosystems characterized by complex human–natural interactions using a modified CASA model coupled with empirical equations for soil respiration, thereby enhancing estimation accuracy and addressing theoretical gaps in regional carbon cycle research; (2) to systematically analyze spatiotemporal vegetation carbon sink dynamics and their multi-scale drivers through the integration of trend analysis, partial correlation analysis, multiple correlation analysis, and the optimal parameter-based GeoDetector. These advancements aim to quantify interactions between climatic drivers (e.g., temperature, precipitation) and anthropogenic pressures (e.g., land use intensity, nighttime light) via GeoDetector-based interaction decomposition, overcoming the limitations of single-factor attribution analysis in existing studies. We hypothesize that (1) coastal urban clusters will demonstrate significantly lower NEP values compared to southwestern mountainous ecosystems due to asymmetric anthropogenic disturbances; (2) ecological civilization policies implemented post-2018 will significantly accelerate the increase rate of carbon sequestration capacity, particularly in peri-urban transition zones. The results of this study are expected to provide critical insights for optimizing ecosystem management strategies and terrestrial ecosystem conservation in rapidly developing coastal regions.

## 2. Materials and Methods

### 2.1. Study Area

Zhejiang Province (118°01′–123°10′ E, 27°02′–31°11′ N), located on the eastern coast of China, is an integral part of the southern wing of the Yangtze River Delta with the area of approximately 105,500 km^2^ [25]. The subtropical monsoon climate endows the province with mean annual precipitation ranging at 1000–2000 mm and mean annual temperature ranging at 15–18 °C [26]. This climatic regime supports 2300–2600 annual sunshine hours and sustains China’s highest recorded forest coverage rate (61.27% as of 2023), predominantly comprising evergreen broad-leaved forests (Fagaceae, Lauraceae) clustered in southwestern highlands [27]. The terrain exhibits a distinct southwest-to-northeast topographic gradient, with elevated mountainous regions (average elevation: 754–1830 m) dominating the southwestern interior, transitioning through intermediate hilly basins to low-lying coastal plains (average elevation: <50 m) in the northeastern deltaic zones. This topographic pattern is consistent with the classic ’seven mountains, one water, two fields’ geomorphic pattern, reflecting the province’s tectonic uplift history and fluvial erosion processes [28] (Figure 1).

### 2.2. Data Sources and Preprocessing

In this study, temperature and precipitation data were downscaled to a high-resolution spatial scale across China using the Delta spatial downscaling method. This process integrated the global 0.5° climate dataset from the CRU (Climatic Research Unit) and the high-resolution global climate dataset from WorldClim [29]. Radiation data were obtained from the TerraClimate dataset, a high-resolution global dataset of the monthly climate and climatic water balance for terrestrial surfaces, covering the period from 1958 to the present [30]; vegetation type data from the MCD12Q1 data product (National Aeronautics and Space Administration, NASA), which incorporates extensive calibration using ground-truth data and machine learning algorithms, ensuring its robustness for regional-scale analyses [31]; and NDVI data from the MOD13Q1 data product (NASA), which were used to simulate NPP values by the CASA model, with a time span from 2001 to 2023 [32]. NPP data of the MOD17A3 data product (NASA) were used to evaluate the accuracy of the simulation results in this paper. DEM data were obtained from the ASTER GDEM30M products available on the Geospatial Data Cloud (https://www.gscloud.cn/). Slope was calculated using the slope analysis tool in ArcGIS 10.8 software.

The land use data and population density data were derived from the Resource and Environmental Science and Data Center of the Chinese Academy of Sciences. The land cover classification system includes six primary categories: arable land, forest, grassland, water, construction land, and unused land. The nighttime light data were obtained from the global 500 m resolution “Class NPP-VIIRS” nighttime light dataset (2000–2022) released by the National Earth System Science Data Center, which is characterized by high spatiotemporal consistency and accuracy. The human footprint and impervious surface data were provided by the Urban Environmental Monitoring and Modeling (UEMM) team at the College of Land Science and Technology, China Agricultural University. The arable land expansion data were sourced from the high-resolution global arable land expansion dataset for the 21st century at 30 m resolution, which was locally calibrated using a large number of training data to ensure high accuracy.

All data (Table 1) were clipped and projected to the same projection coordinates by ArcGIS, and bilinear interpolation was used to resample the spatial resolution to 1 km to ensure the consistency of data resolution.

### 2.3. Methods

#### 2.3.1. Improved-CASA Model and the Accuracy Validation of NPP

This study utilized precipitation, temperature, solar radiation, NDVI data, and vegetation type data from 2000 to 2023 to estimate NPP in Zhejiang Province using the improved CASA model developed by Zhu et al. (2023) [33]. The improved CASA model incorporated advanced methodologies to enhance the estimation of key drivers of NPP, including the fraction of absorbed photosynthetically active radiation, water stress, and temperature stress coefficients. These refinements significantly enhance the accuracy of NPP calculations. Meanwhile, this model enables dynamic coupling of multi-factor drivers (climate–anthropogenic gradients) and demonstrates strong applicability to complex ecosystems, thereby providing a robust methodological tool for carbon sink assessments in rapidly urbanizing regions [33]. The calculation formula is as follows:NPP(x, t) = APAR(x, t) × ε(x, t)

In the equation, x: spatial location, in pixels; t: time, in months; NPP(x, t): Net Primary Productivity of pixel x at time t, gC·m^−2^·a^−1^; APAR(x, t): Photosynthetically Active Radiation of pixel x at time t, MJ·m^2^·a^−1^; ε(x, t): light use efficiency (LUE) of vegetation for pixel x at time t, gC·MJ^−1^.

Given the significant uncertainties in estimating NPP at regional or global scales, validation of accuracy is necessary. This study validated the accuracy of NPP estimates by comparing them with the MODIS NPP dataset (MOD17A3) [34]. Within the study area, we randomly selected 100 sample points and extracted the average values of both the estimated and reference NPP data at these locations. Correlation analysis and significance testing were then conducted on these values. The estimated NPP exhibited a significant linear correlation with the MODIS NPP time series (Figure 2), with a correlation coefficient of 0.90 (*p* < 0.05, R^2^ = 0.805, n = 100). This result confirmed the high reliability of our NPP estimates in Zhejiang Province.

#### 2.3.2. The Estimation Model of Vegetation NEP

NEP is defined as the difference between the NPP and the carbon emissions from Rh within an ecological zone [3]. It is considered the rate of carbon exchange between terrestrial and atmospheric ecosystems and serves as a crucial indicator for estimating the regional carbon balance, often used as a measure of the carbon sink capacity [12]. The formula for calculating NEP isNEP (x, t) =NPP (x, t) − Rh (x, t)

In the equation, x: spatial location, in pixels; t: time, in months; NEP(x, t): Net Ecosystem Productivity of pixel x at time t, gC·m^−2^·a^−1^; NPP(x, t): Net Primary Productivity of pixel x at time t, gC·m^−2^·a^−1^; Rh: soil heterotrophic respiration, gC·m^−2^·a^−1^. When NEP > 0, the ecosystem acts as a carbon sink; conversely, when NEP < 0, it acts as a carbon source [6].Rh (x, t) =0.22 × (e^0.0913T^ + ln(0.3145R) + 1)) × 30 × 46.5%
where T is the air temperature (°C) and R is the precipitation (mm).

#### 2.3.3. Trend Analysis Method

Based on the univariate linear regression model using the least squares method [35], this study conducted a time-series analysis of the trends in NEP in Zhejiang Province from 2000 to 2023, with the trend rate used to characterize the changes. The formula is as follows:$$\mathrm{Slope}=\frac{n\times \sum_{i=1}^{n}i\times {NEP}_{i}-(\sum_{i=1}^{n}i)(\sum_{i=1}^{n}{NEP}_{i})}{n\times \sum_{i=1}^{n}i^{2}-({\sum_{i=1}^{n}i)}^{2}}$$

In the equation, Slope: the trend of NEP changes; *n*: the study period, *n* = 23; *NEP_i_*: the average NEP for the *i*-th year. In the calculation results, Slope > 0 indicates an upward trend in NEP, while Slope < 0 indicates a downward trend.

The F-test was used to test the significance of the NEP trend, categorizing the NEP changes into five levels: extremely significant increase (*p* < 0.01, Slope > 0), significant increase (0.01 < *p* < 0.05, Slope > 0), stable (*p* > 0.05), significant decrease (0.01 < *p* < 0.05, Slope < 0), and extremely significant decrease (*p* < 0.01, Slope < 0).

#### 2.3.4. Partial and Multiple Correlation Analyses

Partial correlation analysis was used to examine the linear relationship between two variables while controlling for the linear effects of other variables. When the number of control variables is two, it is referred to as the second-order partial correlation coefficient [36]. Multiple correlation analysis was employed to examine the collective impact of various climatic factors on NEP and to analyze their interrelationships [37]. In this study, Python 3.9 was utilized to perform pixel-by-pixel analysis of the second-order partial correlations and multiple correlations between NEP and temperature, precipitation, and solar radiation.

The significance of the partial correlation coefficients and multiple correlation coefficients was tested using T-tests and F-tests, respectively. The coefficients were categorized as follows: highly significant (*p* < 0.01), significant (0.01 < *p* < 0.05), and not significant (*p* > 0.05). To quantify the contribution of climatic factors to NEP changes, this study established nine climatic driving types based on existing classification criteria. The specific types were as follows: temperature-driven, precipitation-driven, solar radiation-driven, temperature and precipitation-driven, temperature and solar radiation-driven, precipitation and solar radiation-driven, strongly driven by temperature, precipitation, and solar radiation, weakly driven by temperature, precipitation, and solar radiation, and non-climatic factor-driven.

#### 2.3.5. GeoDetector Based on Optimal Parameters (OPGD)

Building upon the traditional GeoDetector, the optimal parameter-based GeoDetector (OPGD) enhanced the model’s accuracy and explanatory power by incorporating parameter optimization mechanisms and multi-factor analysis, thereby revealing spatial relationships among variables. In this study, four detectors (Factor Detector, Optimal Parameter Detector, Interaction Detector, and Risk Detector) were selected based on the GD package in R language [38]. These detectors were used to perform optimal parameter analysis on factors requiring discretization, aiming to determine the optimal categories of influencing factors. The explanatory power of independent variables on the dependent variable was measured, and the relationships between explanatory factors and the analyzed variables were interpreted using the q-value [39]. The q-value ranges as [0, 1], where a higher value indicates a stronger explanatory ability of the factor on the spatial heterogeneity of the dependent variable (e.g., vegetation NEP). The formula for calculating the q-value is as follows:$$q=1-\frac{\sum_{h=1}^{L}N_{h}\times \sigma_{h}^{2}}{N\times {\;\sigma }^{2}}$$
where *N_h_*: the number of samples in the *h*-th stratum; $\sigma_{h}^{2}$: the variance within the *h*-th stratum; *N*: the total number of samples; $\sigma^{2}$: the total variance; *L*: the number of strata.

The function of interaction detection was applied to identify the interaction effects between different factors, specifically to evaluate whether the combined influence of factors X1 and X2 enhanced or diminished the explanatory power on the dependent variable Y. The types of interaction effects included linear enhancement, nonlinear enhancement, independence, linear weakening, and nonlinear weakening.

In this study, 12 factors were selected as detection factors (Table 2). Among these, impervious surface change (3 categories), human footprint change (6 categories), and land use change (12 categories) were classified according to the original data standards. The remaining factors were classified based on their multi-year averages using optimal parameters. The dependent variable of NEP was also calculated as the multi-year average.

## 3. Results

### 3.1. Temporal and Spatial Evolution of NPP in Zhejiang Province

The annual mean NPP in Zhejiang Province from 2000 to 2023 ranged between 644.2 g C m^−2^ a^−1^ and 830.08 g C m^−2^ a^−1^, with a multi-year average of 731.61 g C m^−2^ a^−1^. The lowest value occurred in 2022, while the peak was observed in 2021. The NPP exhibited a fluctuating upward trend, with an average annual increase of 3.011 g C m^−2^ a^−1^ (Figure 3b).

Spatially, NPP displayed an uneven distribution from 2001 to 2023, characterized by higher values in the western and southern regions and lower values in the eastern and northern areas. High NPP zones (>1200 g C m^−2^ a^−1^) were concentrated in the mountainous regions of Lishui and western Wenzhou, which accounted for 6.464% of the total area. In contrast, low NPP zones (0–600 g C m^−2^ a^−1^) dominated the Hangzhou Bay coastal areas and other coastal regions, covering 24.281% of the province. Intermediate values (600–900 g C m^−2^ a^−1^) represented the largest proportion (63.973%) (Figure 3a).

The significance of the change trend of NPP from 2000 to 2023 in Zhejiang Province showed clear spatial variations, and NPP showed an overall increasing trend. The result revealed that 47.82% of the province exhibited a significant increase in NPP (*p* < 0.05), widely distributed across the region. Only 0.75% of areas (primarily coastal zones) showed a significant decline, while 51.43% remained stable (Figure 3c,d).

### 3.2. Temporal and Spatial Evolution of Vegetation Carbon Sink (NEP) in Zhejiang Province

#### 3.2.1. Temporal Evolution of Vegetation Carbon Sink

The annual mean NEP in Zhejiang Province from 2000 to 2023 ranged from 282.66 g C m^−2^ a^−1^ to 469.65 g C m^−2^ a^−1^, with a multi-year average of 387.67 g C m^−2^ a^−1^. Similar to NPP, the lowest NEP occurred in 2022, and the highest value was recorded in 2021. The trend showed a fluctuating increase at an average rate of 2.15 g C m^−2^ a^−1^ (Figure 4a).

In terms of total carbon sequestration, the multi-year average carbon sink capacity was 38.84 Tg C a^−1^, while carbon sources contributed minimally (−0.40 Tg C a^−1^). Both carbon sinks and sources exhibited slight upward trends, with annual increases of 0.23 Tg C a^−1^ and 0.01 Tg C a^−1^, respectively. These results indicated that Zhejiang Province functioned predominantly as a carbon sink, with its sequestration capacity gradually strengthening over time (Figure 4b).

#### 3.2.2. Spatial Evolution of Vegetation Carbon Sink

Spatially, NEP was higher in the western and southern regions, whereas lower values dominated the eastern coastal and northern plains (Figure 5a,b). During the study period, 96.73% of the province acted as carbon sinks, while carbon sources (3.27%) were concentrated in coastal urban zones (e.g., Jiaxing, eastern Huzhou).

The significance of the change trend of NEP from 2000 to 2023 in Zhejiang Province showed clear spatial variations. The results revealed that 36.75% of areas exhibited a significant increase in NEP (*p* < 0.05), distributed broadly across the province. Conversely, 4.69% of regions (notably northeastern coastal and scattered central areas) showed significant declines, while 58.56% remained stable (Figure 5c,d).

### 3.3. Driving Factors of Vegetation Carbon Sink

#### 3.3.1. Effects of Meteorological Factors on Vegetation NEP

NEP exhibited a positive correlation with temperature, precipitation, and solar radiation across the study region. Specifically, temperature exhibited positive correlations in 64.41% of the region, with 9.05% showing statistically significant positive associations (*p* < 0.05), concentrated in eastern coastal zones (e.g., Hangzhou Bay) and urbanized areas. Precipitation displayed positive correlations in 78.11% of the area, including 14.98% with strong significance, predominantly in northern Zhejiang (e.g., Jiaxing and Huzhou). Similarly, 79.89% of the region showed positive correlations with solar radiation, with 9.98% reaching statistical significance, widely distributed across the province. These findings indicated that temperature, precipitation, and solar radiation primarily acted in a coupled manner to jointly control NEP accumulation in these regions (Figure 6a).

Spatial analysis of multiple correlation coefficients revealed pronounced heterogeneity, with significant positive correlations (11.22% of the total area) clustered in northeastern (e.g., Shaoxing) and northwestern regions (e.g., Jiaxing, Huzhou) (Figure 6b). Meteorological factors collectively accounted for 14.67% of observed NEP variability, with tripartite interactions (temperature–precipitation–radiation) dominating 6.54% of the area and precipitation dominating 5.49% of the area. This underscored the climate’s pivotal role in regulating carbon sequestration dynamics (Figure 6c).

#### 3.3.2. Effects of Anthropogenic and Natural Factors on Spatial Differentiation of Vegetation NEP

All 12 investigated factors significantly influenced the spatiotemporal heterogeneity of NEP (*p* < 0.01). Driver contributions ranked in descending order as NDVI > elevation > nighttime light intensity > slope > impervious surface dynamics > precipitation > land use change > population density > temperature > solar radiation > arable land expansion. NDVI (q = 0.502) and elevation (q = 0.373) emerged as primary natural drivers, while anthropogenic factors, particularly nighttime light intensity (q = 0.327), impervious surface dynamics (q = 0.295), land use change (q = 0.241), and population density (q = 0.233), showed substantial human-induced impacts (Figure 7b).

Factor interaction analysis revealed nonlinear enhancement effects, with paired factors consistently outperforming individual drivers (Figure 7a). NDVI demonstrated particularly strong synergies, with all interaction q-values exceeding 0.50, among which NDVI–precipitation interaction achieved the maximal explanatory power (q = 0.523). Notably, although the single-factor influence of annual human footprint change and arable land expansion change was relatively low, their interactions with other factors all showed a nonlinear enhancement combination with an explanatory power greater than 20%, which was far higher than the sum of single factors. This further indicated that the spatiotemporal variability in NEP was the result of the combined effects of natural and anthropogenic factors.

The optimal factor range or type of NEP in Zhejiang Province is influenced by multiple factors. The NEP increased rapidly with the rise in precipitation, NDVI, elevation, and slope. In contrast, it decreased rapidly with the increase in temperature, population density, and nighttime light intensity. Additionally, NEP showed a trend of first increasing and then decreasing with the rise in solar radiation. The highest NEP values were observed when the impervious surface remained unchanged in non-urban areas, the human footprint shifted from a severe to a minor impact, and land use remained unchanged in non-arable land and forest areas (Figure 8, Table 3).

## 4. Discussion

The temporal evolution of NEP in Zhejiang Province from 2000 to 2023 exhibited a fluctuating upward trend, consistent with general patterns observed in other subtropical and Yangtze River Delta regions (e.g., Jiangsu Province) [40]. However, Zhejiang’s NEP demonstrated unique variability in magnitude and annual fluctuations. For instance, Fujian Province, another subtropical monsoon climate region, showed relatively stable NEP values (300–400 g C m^−2^ a^−1^) over the past two decades [41], whereas Zhejiang exhibited a higher mean NEP (387.67 g C m^−2^ a^−1^) and greater interannual variability. This discrepancy likely stems from Zhejiang’s intensive ecological restoration initiatives and forest conservation policies. As a national pilot zone for ecological civilization, the forest coverage in Zhejiang increased from 54.2% in 2000 to 61.27% in 2023, significantly surpassing the national average (23%) and directly enhancing carbon sequestration capacity. The anomalously low NEP in 2022 (282.66 g C m^−2^ a^−1^) coincided with extreme summer heatwaves and drought, mirroring similar short-term carbon sink suppression also observed during the 2022 European heatwave [42]. Statistically, 2022 recorded Zhejiang’s third-highest annual temperature (0.8 °C above average) and a 20% reduction in precipitation during the flood season. This phenomenon highlights the “dual nature” of monsoons in shaping regional carbon sinks: While monsoon climates enhance vegetation productivity through stable precipitation, extreme climatic events can impose significant short-term suppression on carbon sinks by reducing NPP. Similar to the Indian monsoon region, where the synergy of heavy rainfall and high temperatures may amplify carbon sink fluctuations [43], Zhejiang Province mitigated such risks through ecological restoration policies, validating the feasibility of “policy–nature synergy”. Overall, the growth of Zhejiang’s NEP underscores the effectiveness of ecological policies. However, the short-term constraints posed by extreme climates remain non-negligible, serving as a critical warning for carbon neutrality pathways in global counterparts.

Overall, Zhejiang Province functions as a significant carbon sink, with a strong vegetation carbon sequestration capacity. The multi-year average carbon sink capacity was 38.84 Tg C a^−1^, higher than other southeastern coastal provinces such as Jiangxi (35 Tg C a^−1^), Hunan (36 Tg C a^−1^), and Guangdong (28.5 Tg C a^−1^) [44]. This high carbon sequestration capacity and its gradual increase (annual increase of 0.23 Tg C a^−1^) can also be attributed to the increased forest cover (reaching 61.27% in 2023) and the implementation of ecological protection policies (e.g., the “Ecological Province” initiative). In summary, Zhejiang Province’s vegetation carbon sequestration capacity stands out in subtropical regions, primarily due to its high forest cover, diverse vegetation types, and proactive ecological restoration policies.

Spatial heterogeneity of NEP in Zhejiang was pronounced, with high values (>469 g C m^−2^ a^−1^) concentrated in the western and southern mountainous regions (e.g., Lishui and Wenzhou) and low values (<300 g C m^−2^ a^−1^) in eastern coastal zones (e.g., Hangzhou Bay) and northern urban belts. This aligned with global and regional carbon sink dynamics, where topographically complex areas with high vegetation cover and minimal human disturbance exhibited elevated NEP [45]. Similar patterns were observed in subtropical Guangdong Province (higher NEP in northern mountains, lower in the Pearl River Delta) [46] and Japan’s Kyushu Island (higher NEP in forested highlands, lower in urbanized coasts) [47]. Zhejiang’s unique “seven mountains, one water, two fields” topography facilitates localized humid microclimates in southwestern highlands (elevation: 754–1830 m; slope: 30.5–75.6°), promoting evergreen broadleaf forest growth (e.g., Fagaceae, Lauraceae) with high light use efficiency (LUE = 0.502). In contrast, eastern coastal plains, characterized by impervious surface expansion (23.2% coverage) and urban heat island effects, suppressed photosynthesis and vegetation NEP. Meanwhile, the eastern coastal areas are characterized by plantation forests and urban green spaces with lower vegetation cover and limited carbon sequestration potential. This was consistent with the findings of Zhang et al. (2019) [48], who reported that natural forests generally had higher NEP than artificial vegetation in subtropical regions. Despite rapid urbanization, Zhejiang’s carbon source area proportion (3.27%) remained lower than the Pearl River Delta (8.5%) [46], attributed to less intensive industrial land use and proactive urban green space management (e.g., Hangzhou’s West Lake ecological zone). Overall, Zhejiang’s NEP spatial patterns underscored the interplay of topography, vegetation diversity, and human activity gradients.

Meteorological factors (temperature, precipitation, solar radiation) significantly influenced vegetation NEP, with positive correlations across 64.41–79.89% of the study area. Tripartite interactions accounted for 6.54% of NEP spatial heterogeneity, consistent with tropical/subtropical regions where hydrothermal synergy optimized carbon sequestration [49]. Precipitation dominated climatic contributions (q = 0.373), exceeding temperature (q = 0.20) and radiation (q = 0.18), suggesting that although temperature, precipitation, and solar radiation had a synergistic effect on vegetation NEP, precipitation played a stronger role. This was consistent with the precipitation dependency of Zhejiang Province’s subtropical monsoon climate. Abundant precipitation (1000–2000 mm/year) alleviated summer droughts and maintained vegetation photosynthetic efficiency, especially in the northern plains (e.g., Jiaxing and Huzhou). However, coastal urban zones (e.g., Hangzhou Bay) experienced NEP declines under elevated temperatures, where heat-induced soil respiration (Rh) and evapotranspiration exacerbated carbon loss, echoing threshold effects observed in India’s monsoon regions [43] and indicating that temperature had a threshold effect on carbon sequestration. In summary, the combined effect of temperature, precipitation, and solar radiation significantly drove NEP in Zhejiang Province, in which precipitation emerged as the primary climatic regulator, though temperature extremes locally counteracted carbon gains.

The pattern of NEP in Zhejiang Province was co-driven by both natural and human factors, with NDVI (q = 0.502) and elevation (q = 0.373) being the primary natural drivers, and nighttime light intensity (q = 0.327) and impervious surface dynamics (q = 0.295) being the main anthropogenic drivers. Regarding natural factors, the high contribution of NDVI was directly related to vegetation cover, supporting the classic theory that “vegetation productivity determines carbon sequestration” [50]. However, the NDVI–precipitation interaction (q = 0.523) revealed nonlinear hydrothermal coupling, enhancing carbon sequestration beyond individual factor effects. Anthropogenically, nighttime light intensity and impervious surface dynamics dominated human impacts, reflecting urbanization-driven carbon sink suppression via habitat fragmentation and albedo alteration [51]. This was consistent with the findings of Ma et al. (2023) [51], who reported that the expansion of impervious surfaces reduced the carbon sequestration potential by decreasing vegetation cover and altering surface albedo in the Yangtze River Delta urban agglomerations. Unlike the Pearl River Delta, where industrial expansion dominated (q > 40%) [44], Zhejiang’s anthropogenic drivers were centered on urban sprawl, and urban green space management (e.g., the West Lake scenic area in Hangzhou) partially offset the negative impact of impervious surfaces. The nonlinear enhancement of factor interactions (e.g., NDVI–population density, elevation–land use change) was common, with q-values exceeding the sum of individual factors, indicating that traditional single-factor analyses may have underestimated the comprehensive effects. In summary, NEP dynamics in Zhejiang Province were co-driven by natural and human factors (e.g., policy-moderated urbanization), highlighting that ecological restoration measures under policy guidance partially offset carbon sequestration losses.

Future research should address the limitations of this study. Although 1 km resolution data were used, the micro-scale heterogeneity in rapidly urbanizing areas (e.g., fragmented urban green spaces) may not have been fully captured. Future studies could combine drone or hyperspectral remote sensing to deepen the analysis. Additionally, while the current study covers 2000–2023, the short-term impact of extreme climate events (e.g., the 2022 heatwave and drought) on carbon sequestration requires validation over a longer time series. As a pilot “Ecological Province”, the specific effects of Zhejiang Province’s reforestation and ecological compensation policies need to be further quantified in combination with socioeconomic data.

## 5. Conclusions

This study provides a comprehensive analysis of the spatiotemporal dynamics of NEP in Zhejiang Province from 2000 to 2023 and identifies the key factors influencing these dynamics. The results showed that Zhejiang Province experienced a gradual increase in vegetation NEP, with the multi-year average of 387.67 g C m^−2^ a^−1^, ranging from 282.66 to 469.65 g C m^−2^ a^−1^. The province predominantly functioned as a carbon sink, with a multi-year average carbon sink capacity of 38.84 Tg C a^−1^, and its sequestration capacity was gradually strengthened over time. The spatial distribution of NEP was characterized by higher values in the western and southern mountainous regions and lower values in the eastern coastal and northern plains. The pattern of NEP in Zhejiang Province was co-driven by both natural and human factors, with NDVI (q = 0.502) and elevation (q = 0.373) being the primary natural drivers, and nighttime light intensity (q = 0.327) and impervious surface dynamics (q = 0.295) being the main anthropogenic drivers.

This spatiotemporal heterogeneity reflected nonlinear interactions between natural factors and human activities. The superior carbon sink capacity of western mountainous areas was attributed to favorable hydrothermal conditions shaped by topography and vegetation protection policies, whereas carbon source zones in eastern coastal regions were dominated by urbanization-induced habitat degradation and climatic stressors. Compared to other rapidly urbanizing regions, Zhejiang exhibited a lower proportion of carbon source areas (3.27%) and moderate human-driven pressures, offering a model for balancing ecological conservation and economic development in global subtropical zones. Future research should integrate high-resolution remote-sensing data to quantify the long-term impacts of extreme climate events and ecological policies, optimize restoration strategies, and advance the “Dual Carbon” goals (carbon peaking and neutrality).

## Figures and Tables

**Figure 1 plants-14-01151-f001:**
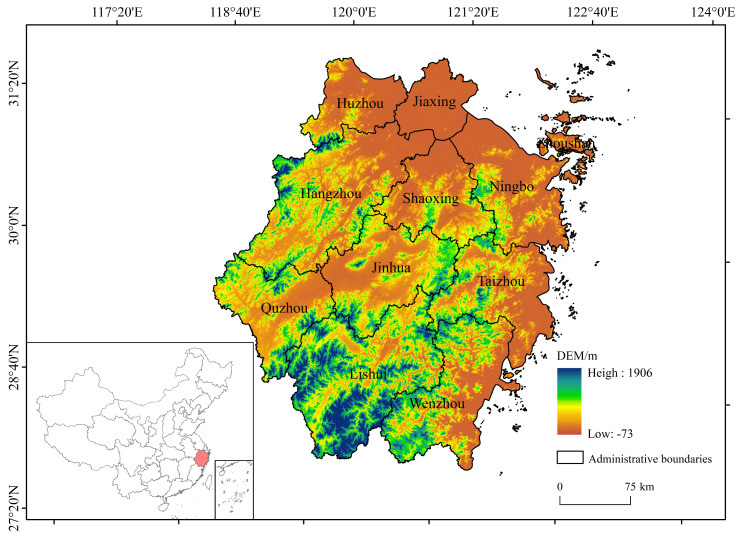
Sketch map of the study area: lower-left corner—Zhejiang Province’s location in China, central part—the elevation of Zhejiang Province.

**Figure 2 plants-14-01151-f002:**
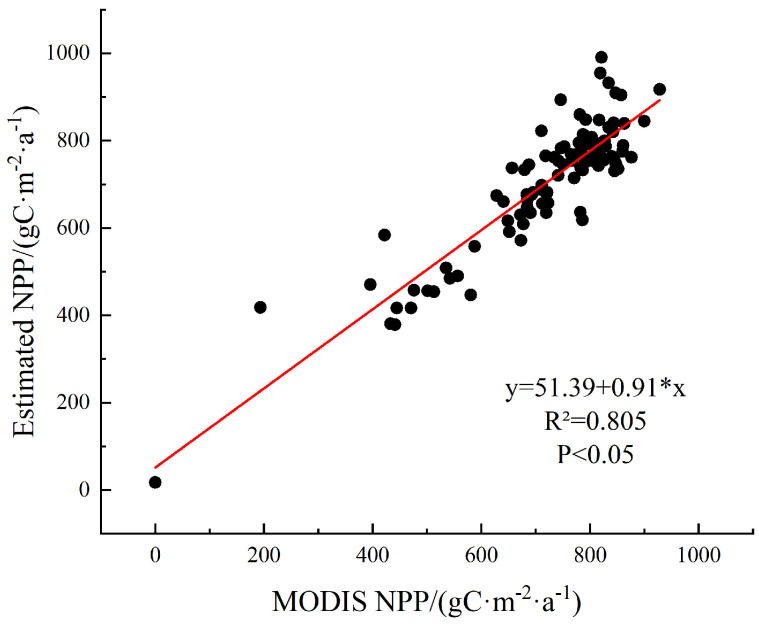
Accuracy validation of NPP.

**Figure 3 plants-14-01151-f003:**
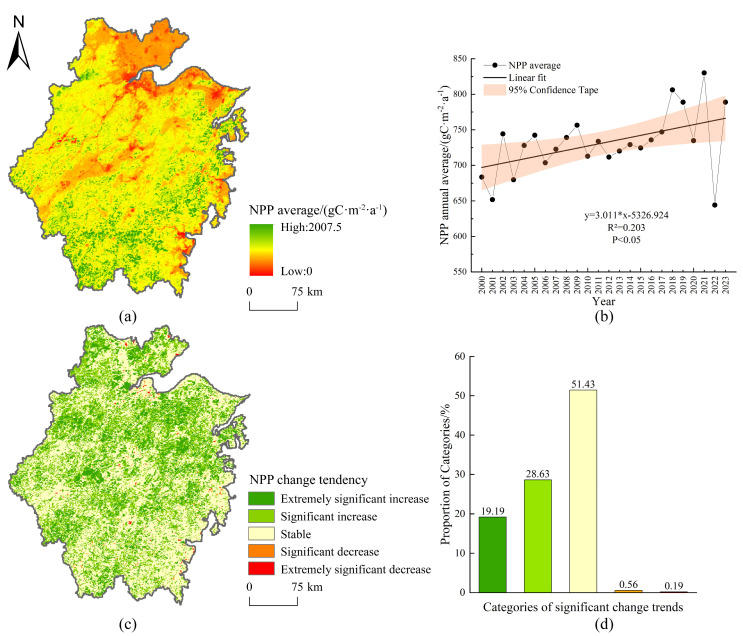
Spatial distribution, temporal variation, and significance of NPP change trends in Zhejiang Province. (**a**) The annual mean NPP from 2000 to 2023; (**b**) Interannual variation trend of average NPP from 2000 to 2023; (**c**) Spatial distribution of the significance of NPP change trends from 2000 to 2023; (**d**) The proportion of the significance of NPP change trends.

**Figure 4 plants-14-01151-f004:**
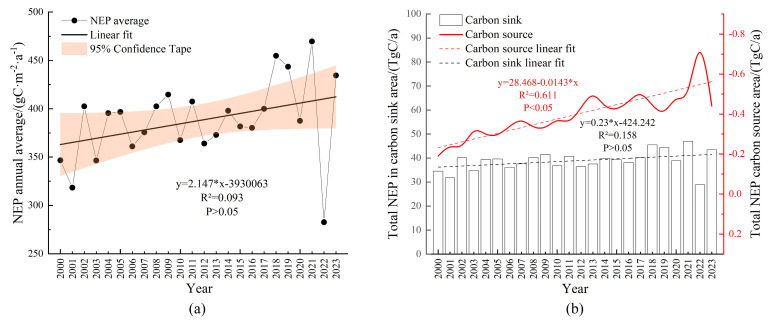
Temporal trends of NEP and total carbon sequestration (sinks/sources). (**a**) Interannual variation trend of average NEP from 2000 to 2023; (**b**) temporal variations in total carbon sequestration in carbon source/sink areas of Zhejiang Province from 2000 to 2023.

**Figure 5 plants-14-01151-f005:**
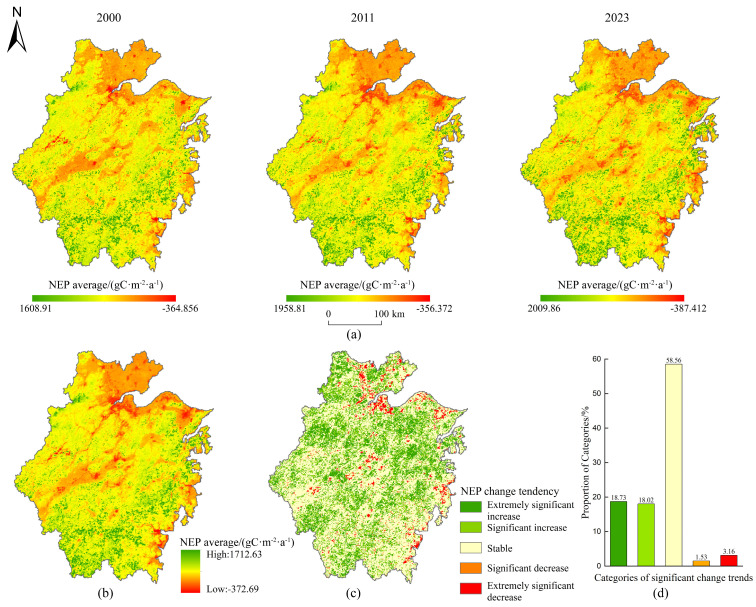
Spatial distribution, temporal variation, and significance of NEP change trends in Zhejiang Province. (**a**) The annual average NEP in the years 2000, 2011, and 2023; (**b**) the annual mean NEP from 2000 to 2023; (**c**) spatial distribution of the significance of NEP change trends from 2000 to 2023; (**d**) the proportion of the significance of NEP change trends.

**Figure 6 plants-14-01151-f006:**
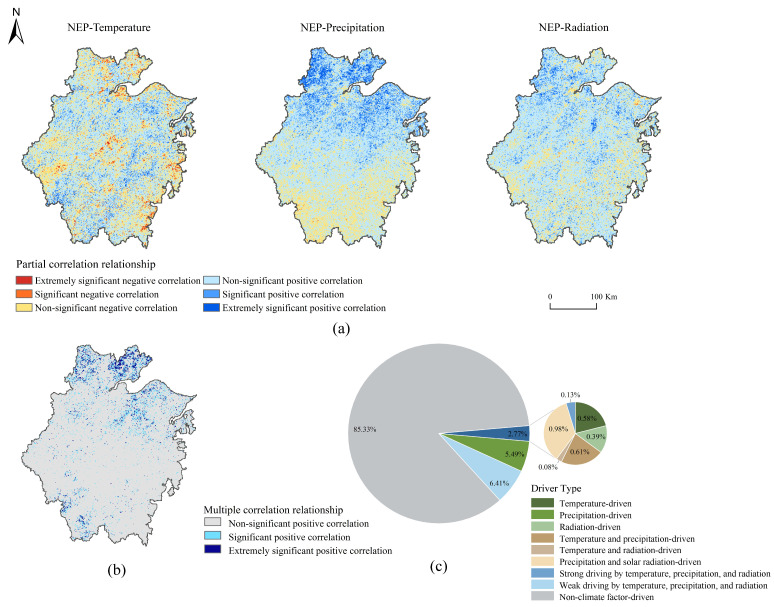
Spatial correlation between NEP and climatic driver contributions in Zhejiang Province. (**a**) Spatial analysis of partial correlations between temperature, precipitation, solar radiation, and NEP; (**b**) spatial analysis of multiple correlations of climatic factors; (**c**) proportions of climate-driven types of NEP.

**Figure 7 plants-14-01151-f007:**
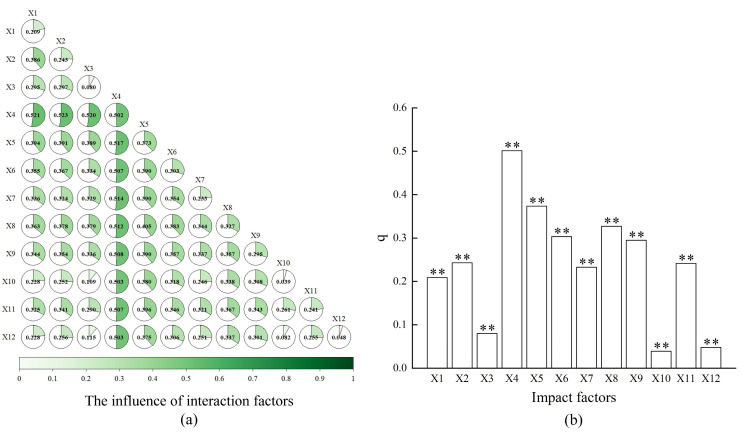
Factor interaction effects and explanatory power rankings of single factors. (**a**) Factor interaction effects; (**b**) explanatory power rankings of single factors. The “**” indicates that the result has passed the significance test and represents an extremely significant impact.

**Figure 8 plants-14-01151-f008:**
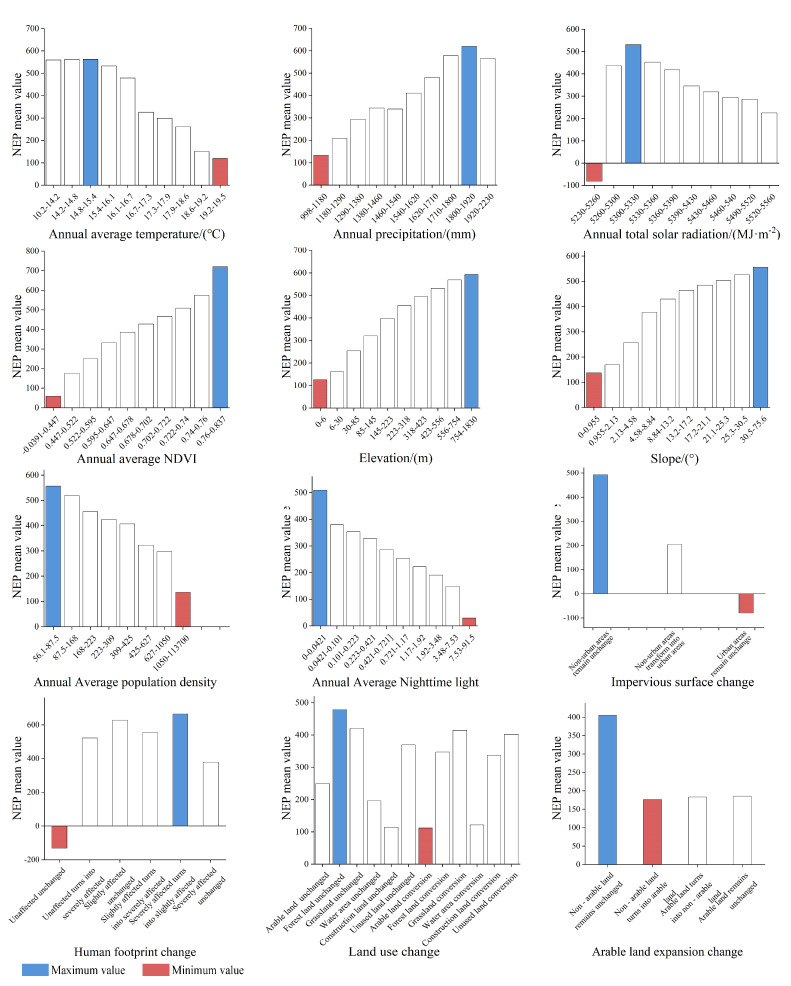
The suitable range and type of NEP for each impact factor. Blue: the most suitable range or type (the maximum NEP); red: the least suitable range or type (the minimum NEP).

**Table 1 plants-14-01151-t001:** Data types and sources.

Data Type	Resolution	Time Span	Data Sources
Temperature	0.5 °C	2000–2022	https://doi.org/10.11888/meteoro.tpdc.270961 (accessed on 6 February 2025) https://cstr.cn/18406.11.meteoro.tpdc.270961 (accessed on 6 February 2025)
Precipitation	0.5°	2000–2022	https://doi.org/10.5281/zenodo.3185722 (accessed on 6 February 2025)
Solar Radiation	4 km	2000–2022	http://thredds.northwestknowledge.net:8080/thredds/catalog/TERRACLIMATE_ALL/data/catalog.html (accessed on 6 February 2025)
Vegetation Type	500 m	2000–2022	https://lpdaac.usgs.gov/products/mcd12q1v006/ (accessed on 6 February 2025)
NDVI	1 km	2000–2022	https://lpdaac.usgs.gov/products/mod13a2v006/ (accessed on 6 February 2025)
Elevation	30 m	-	https://www.gscloud.cn/ (accessed on 6 February 2025)
Slope	30 m	-	-
Land Use	300 m	2000, 2020	https://www.resdc.cn/DOI/DOI.aspx?DOIID=54 (accessed on 6 February 2025)
Population Density	1 km	2000, 2010, 2020	https://www.resdc.cn/DOI/DOI.aspx?DOIID=32 (accessed on 6 February 2025)
Nighttime Light	500 m	2000–2020	http://www.geodata.cn (accessed on 6 February 2025)
Human Footprint	1 km	2000–2020	https://www.x-mol.com/groups/li_xuecao/news/48145 (accessed on 6 February 2025)
Impervious Surface	30 m	2000–2020	-
Arable land Expansion	30 m	2000–2019	https://glad.umd.edu/dataset/croplands (accessed on 6 February 2025)

**Table 2 plants-14-01151-t002:** Impact factors and classification.

Indicator Category	Impact Factors	Number	Discretization Classification
Natural factor	**Annual Average Temperature**	X1	10
	**Annual Precipitation**	X2	10
	**Annual Total Solar Radiation**	X3	10
	**Annual Average NDVI**	X4	10
	Elevation	X5	10
	Slope	X6	10
Human factor	**Annual** Average Population Density	X7	8
	**Annual** Average Nighttime Light	X8	10
	Impervious Surface Change	X9	3
	Human Footprint Change	X10	6
	Land Use Change	X11	12
	Arable Land Expansion Change	X12	4

**Table 3 plants-14-01151-t003:** The optimal factor range or type of vegetation NEP.

Indicator Category	Impact Factors	Range or Type of Suitability	Vegetation NEP/(g C m^−2^ a^−1^)
Natural factor	Annual Average Temperature (X1)	14.8–15.4/℃	562.55
	Annual Precipitation (X2)	1800–1920/mm	619.68
	Annual Total Solar Radiation (X3)	5300–5330/(MJ/m^2^)	530.53
	Annual Average NDVI (X4)	0.76–0.837	720.21
	Elevation (X5)	754–1830/m	592.64
	Slope (X6)	30.5–75.6/°	556.03
Human factor	Annual Average Population Density (X7)	56.1–87.5/(Person/km^2^)	556.11
	Annual Average Nighttime Light (X8)	0–0.0421	509.08
	Impervious Surface Change (X9)	Unchanged in non-urban areas	492.78
	Human Footprint Change (X10)	Severe to minor impact areas	663.68
	Land Use Change (X11)	Unchanged in forest areas	479.25
	Arable Land Expansion Change (X12)	Unchanged in non-arable land areas	405.49

## Data Availability

Data is contained within the article.

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
