# Peer review of "Spatiotemporal Dynamics and Drivers of Vegetation Carbon Sinks in Zhejiang Province: A Case Study in Rapidly Urbanizing Subtropical Ecosystems"

_plants, 2025, doi:10.3390/plants14071151_

Round 1

Reviewer 1 Report

Comments and Suggestions for Authors

Spatiotemporal Dynamics and Drivers of Vegetation Carbon Sinks in Zhejiang Province: A Case Study in Rapidly Urbanizing Subtropical Ecosystems

The evaluation of NEP (net ecosystem productivity) represents an interesting goal, especially nowadays under an increase of CO2 in the atmosphere.

In this study the authors aim to estimate the net ecosystem productivity in the Zhejiang Province, in China, combining different model approaches and they highlighted the importance of combining natural and anthropogenic-induced impacts to evaluate the capacity of the area as a carbon sink and forecast China’s carbon neutrality goals.

The objectives of this studies are ambitious, but this is an important topic and tool for climate control and land management.

The abstract is well written, and the keywords are well defined.

Introduction

Line 48. In this sentence “mapping regional vegetation NEP dynamics”, I think it is better to use the word territory or ecosystem instead of vegetation. Because NEP involves all the ecosystem processes. The same in line 51 and other lines.

Lines 103-112. The objectives of the study should be presented in a more schematic way, probably indicating point by point the different objectives. Additionally, a hypothesis of the expected results should be presented.

Material and methods

Lines 123-124. “The terrain is characterized by a high southwest and low northeast, with a stepped gradient from southwest to northeast”. The sentence is not totally clear and should need a better explanation.

The authors should present a clear explanation in the legend of figure 1.

The explanation of the mathematical formulas and models is clear and well addressed.

Results

Figure 3. Although it is explained in the manuscript, the legend of the figure should contain a better explanation. Additionally, the legends of the maps are extremely small and difficult to read.

Figure 4b is difficult to understand and it should have a better explanation in figure legend.

Figure 5. The legend of this figure should contain a better explanation. The authors should explain the different maps and the graphic of the trends.

Figure 6. In the same way the legend should have a better explanation.

In my opinion figure 8 is difficult to understand and needs a better explanation in the text and figure legend.

The discussion and conclusions are well explained.

I think the manuscript represents a valuable work, and it presents an interesting study to quantify the NEP in this region, the temporal and spatial scales and the drivers for change.

In my opinion it deserves to be published in the journal, but it needs to improve the objectives and all the graphics should be more clear and, especially, the figure legends should have a better explanation in order the readers could understand the information without reading all the text.

Author Response

Introduction

Line 48. In this sentence “mapping regional vegetation NEP dynamics”, I think it is better to use the word territory or ecosystem instead of vegetation. Because NEP involves all the ecosystem processes. The same in line 51 and other lines.

Response: Thank you for your valuable suggestion. Combined with the suggestion (For consistency, add “vegetation” before all NEP mentions throughout the text, og rename the abbreviation) of the reviewer 2, NEP reflects whether the entire ecosystem (vegetation + soil + microbes, etc.) is a carbon sink (positive) or carbon source (negative), calculated as the difference between net primary productivity (NPP) and soil heterotrophic respiration (Râ‚•). Since the E of NEP represents ecosystem, we will no longer add the word “ecosystem or territory” in front of NEP, and the full text will be changed to “NEP” instead of “vegetation NEP” to avoid ambiguity.

Lines 103-112. The objectives of the study should be presented in a more schematic way, probably indicating point by point the different objectives. Additionally, a hypothesis of the expected results should be presented.

Response: Thanks for your insightful comments.

We have revised the text to explicitly link the objectives to the identified gaps in Line 109-121. Therefore, this study focuses on the rapidly urbanizing subtropical ecosystem in Zhejiang Province and aims to achieve several key objectives: (1) to estimate Net Ecosystem Productivity (NEP) in heterogeneous ecosystems characterized by complex human-natural interactions using a modified CASA model coupled with empirical equations for soil respiration, thereby enhancing estimation accuracy and addressing theoretical gaps in regional carbon cycle research; (2) to systematically analyze spatiotemporal vegetation carbon sink dynamics and their multi-scale drivers through the integration of trend analysis, partial correlation analysis, multiple correlation analysis, and optimal parameter-based GeoDetector. These advancements aim to quantify interactions between climatic drivers (e.g., temperature, precipitation) and anthropogenic pressures (e.g., land-use intensity, nighttime light) via GeoDetector-based interaction decomposition, overcoming the limitations of single-factor attribution analysis in existing studies.

Additionally, a hypothesis of the expected results have been presented in Line 121-130. We hypothesize that: (1) Coastal urban clusters exhibit significantly lower NEP values compared to southwestern mountainous ecosystems due to asymmetric anthropogenic disturbances; (2) Ecological civilization policies implemented post-2018 will significantly enhance the increase rate of carbon sequestration capacity, particularly in peri-urban transition zones. The results of this study are expected to provide critical insights for optimizing ecosystem management strategies and terrestrial ecosystem conservation in rapidly developing coastal regions.

Material and methods

Lines 123-124. “The terrain is characterized by a high southwest and low northeast, with a stepped gradient from southwest to northeast”. The sentence is not totally clear and should need a better explanation.

Response: We sincerely appreciate the reviewer's constructive feedback. To enhance clarity, we have revised the description of Zhejiang's terrain in Line 142-148: The terrain exhibits a distinct southwest-to-northeast topographic gradient, with elevated mountainous regions (average elevation: 754–1,830 m) dominating the southwestern interior, transitioning through intermediate hilly basins to low-lying coastal plains (average elevation: <50 m) in the northeastern deltaic zones. This topographic pattern is consistent with the classic 'seven mountains, one water, two fields' geomorphic pattern, reflecting the province's tectonic uplift history and fluvial erosion processes.

The authors should present a clear explanation in the legend of figure 1.

Response: Thanks for your insightful comments. We have presented a clear explanation in the legend of figure 1 and also added explanation in Notes (Line 150-151): Fig.1 The sketch map of the study area: the lower-left corner-Zhejiang Province's location in China, the central part-the elevation of Zhejiang Province.

Results

Figure 3. Although it is explained in the manuscript, the legend of the figure should contain a better explanation. Additionally, the legends of the maps are extremely small and difficult to read.

Response: Thanks for your insightful comments. 1)We have added more explanation in Notes in Figure 3 (Line 293-297): Fig.3. Spatial distribution, temporal variation, and significance of NPP change trends in Zhejiang Province. (a) The annual mean NPP from 2000 to 2023; (b) Interannual variation trend of average NPP from 2000 to 2023; (c) Spatial distribution of the significance of NPP change trends from 2000 to 2023; (d) The proportion of the significance of NPP change trends. 2) Additionally, the legends of the maps are changed to bigger.

Figure 4b is difficult to understand and it should have a better explanation in figure legend. 

Response: We are very grateful for your feedback. 1)We’ve included more explanations in the notes(Line 330-333): Fig.4. Temporal trends of NEP and total carbon sequestration (sinks/sources). (a) Interannual variation trend of average NEP from 2000 to 2023; (b) Temporal variations of total carbon sequestration in carbon source/sink areas of Zhejiang Province from 2000 to 2023. 2) In addition, We have added relevant legends to Figure 4 (b) for easier comprehension and readability.

Figure 5. The legend of this figure should contain a better explanation. The authors should explain the different maps and the graphic of the trends.

Response: We truly appreciate your remarks. 1) We’ve added more explanations to the caption of Figure 5(Line 346-350): Fig. 5. Spatial distribution, temporal variation, and significance of NEP change trends in Zhejiang Province. (a) The annual average NEP in 2000, 2011, and 2023; (b) The annual mean NEP from 2000 to 2023; (c) Spatial distribution of the significance of NEP change trends from 2000 to 2023; (d) The proportion of the significance of NEP change trends; 2) In addition, we’ve adjusted the legend size in the figure to make it clearer.

Figure 6. In the same way the legend should have a better explanation.

In my opinion figure 8 is difficult to understand and needs a better explanation in the text and figure legend.&nbsp

Response: Thank you very much for your comments. 1) We’ve added more explanations to the caption of Figure 6(Line 354-357) and Figure 8(Line 383-386): Fig.6. Spatial correlation between NEP and climatic driver contributions in Zhejiang Province. (a) Spatial analysis of partial correlations between temperature, precipitation, solar radiation and NEP; (b) Spatial analysis of multiple correlations of climatic factors; (c). Proportion of climate-driven types of NEP; Fig.8. The suitable range and type of NEP for each impact factor. Blue: the most suitable range or type(the maximum NEP); Red: the least suitable range or type(the minimum NEP).

I think the manuscript represents a valuable work, and it presents an interesting study to quantify the NEP in this region, the temporal and spatial scales and the drivers for change. In my opinion it deserves to be published in the journal, but it needs to improve the objectives and all the graphics should be more clear and, especially, the figure legends should have a better explanation in order the readers could understand the information without reading all the text.

Response: Thank you for recognizing the value of our manuscript and for your thorough and constructive feedback. We have carefully reviewed your suggestions and have improved the objectives, the figure legends have a better explanation and more clear.

Reviewer 2 Report

Comments and Suggestions for Authors

I have reviewed the paper entitled “Spatiotemporal Dynamics and Drivers of Vegetation Carbon 2 Sinks in Zhejiang Province: A Case Study in Rapidly Urbanizing Subtropical Ecosystems” 

The work is well planned and the time period covered is representative. However, it is necessary to justify what this study contributes to global knowledge by looking for interrelations at a global level. The study area is subject to a monsoon period, something that does not occur in other parts of the world (Europe, America, etc.), therefore it is necessary to establish a parallelism based on similar studies.

Has any kind of calibration been carried out on the measurements taken using remote sensors and measurements of the type of vegetation through field evaluations?

Table 1 is cut off. Units of measurement should be separated from the figure (e.g. 30 m, 0.5 °C), please check this in the manuscript

Figure 5 is blurred, please increase the image resolution

Author Response

Comments and Suggestions for Authors I have reviewed the paper entitled “Spatiotemporal Dynamics and Drivers of Vegetation Carbon 2 Sinks in Zhejiang Province: A Case Study in Rapidly Urbanizing Subtropical Ecosystems”  The work is well planned and the time period covered is representative. However, it is necessary to justify what this study contributes to global knowledge by looking for interrelations at a global level. The study area is subject to a monsoon period, something that does not occur in other parts of the world (Europe, America, etc.), therefore it is necessary to establish a parallelism based on similar studies.

Response: We are very grateful for your feedback. We have revised the text to further highlight this study’s global relevance by comparing it with similar parallel studies in other regions in Line 428-438. We have made the following revision in the discussion: This phenomenon highlights the "dual nature" of monsoons in shaping regional carbon sinks: While monsoon climates enhance vegetation productivity through stable precip-itation, extreme climatic events can impose significant short-term suppression on car-bon sinks by reducing NPP. Similar to the Indian monsoon region, where the synergy of heavy rainfall and high temperatures may amplify carbon sink fluctuations, Zhejiang Province has partially mitigated such risks through ecological restoration policies, validating the feasibility of "policy-nature synergy". Overall, the growth of Zhejiang’s NEP underscores the effectiveness of ecological policies. However, the short-term constraints posed by extreme climates remain non-negligible, serving as a critical warning for carbon neutrality pathways in global counterparts. 

Has any kind of calibration been carried out on the measurements taken using remote sensors and measurements of the type of vegetation through field evaluations?

Response: Thanks for your insightful comments.

Yes, calibration and validation procedures were implemented for remote sensing-based measurements in this study. Specifically: 1) NPP Validation with MODIS Data in Line 203-211: The accuracy of the improved CASA model’s NPP estimates was rigorously validated using the MOD17A3 NPP dataset. We randomly selected 100 sample points across Zhejiang Province and performed correlation analysis between our estimated NPP values and the MODIS reference data. The results demonstrated a strong linear correlation (Pearson’s r = 0.90, P < 0.05, R² = 0.805), confirming the high reliability of the model (Section 2.3.1, Fig. 2). 2) Vegetation Type Data: Vegetation type classification relied on the standardized MCD12Q1 product from NASA, which provides globally validated land cover classifications at 500 m resolution. The MCD12Q1 product itself incorporates extensive calibration using ground-truth data and machine learning algorithms, ensuring its robustness for regional-scale analyses. Thus, we have explicitly mentioned the evaluations in Line 159-162. 3) Data Consistency Measures: All datasets (e.g., NDVI, climate variables, land use) were reprojected to a unified coordinate system and resampled to 1 km resolution using bilinear interpolation to minimize spatial mismatches (Section 2.2). This preprocessing step enhanced the coherence of multi-source data integration. The use of validated remote sensing products (e.g., MODIS NPP and MCD12Q1) and cross-validation with independent datasets ensured the methodological rigor of our estimates.

Table 1 is cut off. Units of measurement should be separated from the figure (e.g. 30 m, 0.5 °C), please check this in the manuscript

Response: Thank you very much for your comments. We noted that Table 1(Line 185-186) appeared cut off in the manuscript. We will carefully check and adjust the table format in the final version to ensure all data is fully displayed. We will also separate units of measurement from numerical values as required, formatting them as "30 m", "0.5 °C", etc., to enhance the table's readability and professionalism.

Figure 5 is blurred, please increase the image resolution

Response: Thank you very much for your comments. We apologize for the low resolution of Figure 5. We will replace it with a higher-resolution version (Line 344-345)to ensure clarity and detail.

Reviewer 3 Report

Comments and Suggestions for Authors

The manuscript titled ‘Spatiotemporal Dynamics and Drivers of Vegetation Carbon Sinks in Zhejiang Province: A Case Study in Rapidly Urbanizing Subtropical Ecosystems’ by Xu, et al. presents a comprehensive analysis of the spatiotemporal dynamics and drivers of vegetation carbon sinks in Zhejiang Province, offering valuable insights for carbon cycle research and ecosystem management in rapidly urbanizing subtropical regions. The study employs robust methodologies, including an improved CASA model and Geo-detector, and is based on extensive data sources. The results effectively highlight the complex interplay of natural and anthropogenic factors influencing vegetation NEP. However, the discussion could be strengthened by deeper mechanistic insights and more targeted comparisons with related studies. Addressing these aspects would further enhance the contribution of the manuscript in the field. Overall, with some revisions, this study has the potential to make significant impacts.

Major comments

  1. I suggest the authors carefully review the manuscript for grammatical accuracy, particularly verb tense. For example, use the present tense when stating general facts. Please ensure all abbreviations are introduced in full upon first mention and used consistently throughout the manuscript.
  2. In the Introduction, after stating the research gaps in NEP for complex ecosystems and the difficulties in analyzing multifactor driving mechanisms, the research objectives should be further refined to make them clearer and more directly linked to the identified gaps.
  3. In the Materials and Methods, the improved CASA model is crucial for NPP estimation in this study. Please add technical details to clarify the improvements and their role in enhancing estimation accuracy.

Minor comments

Line 11: What does "CASA" stand for? Please provide the full name of the model if applicable.

Line 12: "Net Ecosystem Productivity" (NEP) should be used in full here.

Lines 26 and 432: change “drived” to “drove”.

Lines 53–54: Ensure that all abbreviations are defined when first introduced.

Line 56: What does "Rh" refer to? Consider rewording for clarity.

Line 60: The sentence should read: ", such as rapidly urbanizing regions, remain…"

Line 63: "Model parameterization" would be a more appropriate term.

Lines 96–106: Consider moving information about Zhejiang to the "Study Area" section.

Lines 131–142: Please add references for the datasets used.

Lines 143–154: It would be beneficial to cite references for the datasets.

Line 156:correct “resampling” to “resample”.

Table 1: A column for references should be included.

Line 171: "Is necessary" should be used instead of "are necessary."

Lines 117-120: have excessive commas. Restructure the sentences for better readability.

Line 182: Since "NPP" and "Rh" were defined in line 44, please use the abbreviations here instead of repeating the full terms.

line 201: The alignment of the Slope formula is inconsistent. Ensure all mathematical

formulas are uniformly aligned and clearly formatted.

Add proper spaces between numerical data and units (e.g., change “30m” to “30 m”,

“4km” to “4 km”, “300m” to “300 m”).

For consistency, add “vegetation” before all NEP mentions throughout the text, og rename the abbreviation.

Comments on the Quality of English Language

I suggest the authors carefully review the manuscript for grammatical accuracy, particularly verb tense. For example, use the present tense when stating general facts. Please ensure all abbreviations are introduced in full upon first mention and used consistently throughout the manuscript.

Author Response

Major comments

Comment 1: I suggest the authors carefully review the manuscript for grammatical accuracy, particularly verb tense. For example, use the present tense when stating general facts. Please ensure all abbreviations are introduced in full upon first mention and used consistently throughout the manuscript.

Response: Thanks for your insightful comments. 1) We have carefully reviewed the entire manuscript to correct grammatical inaccuracies and ensure consistent use of verb tenses. Specifically, all general facts, established theories are now presented in the present tense, and statements describing results are now presented in the past tense. 2) In the revised manuscript, we have systematically ensured that all abbreviations are explicitly defined upon their first mention. Key examples include: Net Ecosystem Productivity (NEP) and Carnegie-Ames-Stanford Approach (CASA) in the revised abstract in Line 11-13, Net Primary Productivity (NPP)(in Line 45 ), soil heterotrophic respiration (Rh)(in Line 46), Moderate Resolution Imaging Spectroradiometer Normalized Difference Vegetation Index (MODIS NDVI), Landsat Thematic Mapper/Operational Land Imager (TM/OLI)) Boreal Ecosystem Productivity Simulator (BEPS) in the revised introduction in Line 55-58.

Comment 2: In the Introduction, after stating the research gaps in NEP for complex ecosystems and the difficulties in analyzing multifactor driving mechanisms, the research objectives should be further refined to make them clearer and more directly linked to the identified gaps.

Response: Thanks for your insightful comments. We have revised the text to explicitly link the objectives to the identified gaps in Line 109-125. Therefore, this study focuses on the rapidly urbanizing subtropical ecosystem in Zhejiang Province and aims to achieve several key objectives: (1) to estimate Net Ecosystem Productivity (NEP) in heterogeneous ecosystems characterized by complex human-natural interactions using a modified CASA model coupled with empirical equations for soil respiration, thereby enhancing estimation accuracy and addressing theoretical gaps in regional carbon cycle research; (2) to systematically analyze spatiotemporal vegetation carbon sink dynamics and their multi-scale drivers through the integration of trend analysis, partial correlation analysis, multiple correlation analysis, and optimal parameter-based GeoDetector. These advancements aim to quantify interactions between climatic drivers (e.g., temperature, precipitation) and anthropogenic pressures (e.g., land-use intensity, nighttime light) via GeoDetector-based interaction decomposition, overcoming the limitations of single-factor attribution analysis in existing studies.

Comment 3: In the Materials and Methods, the improved CASA model is crucial for NPP estimation in this study. Please add technical details to clarify the improvements and their role in enhancing estimation accuracy.

Response: We sincerely appreciate your suggestion to clarify the technical enhancements of the improved CASA model. In the revised Section 2.1 (Materials and Methods), we have added the following details to highlight the methodological advancements and their impact on accuracy in Line 190-197: The improved CASA model incorporates advanced methodologies to improve the estimation of key drivers of NPP, including the fraction of absorbed photosynthetically active radiation, water stress, and temperature stress coefficients. These refinements significantly enhance the accuracy of NPP calculations. Meanwhile, this model enables dynamic coupling of multi-factor drivers (climate-anthropogenic gradients) and demonstrates strong applicability to complex ecosystems, thereby providing a robust methodological tool for carbon sink assessments in rapidly urbanizing regions.

Minor comments

Line 11: What does "CASA" stand for? Please provide the full name of the model if applicable.

Response: Thanks for your insightful comments. The full name with abbreviation of CASA (Carnegie-Ames-Stanford Approach) model has been explicitly defined upon its first mention in the revised abstract in Line 11-12.

Line 12: "Net Ecosystem Productivity" (NEP) should be used in full here.

Response: Thanks for your insightful comments. The full name with abbreviation of NEP (Net Ecosystem Productivity) model has been explicitly defined upon its first mention in the revised abstract in Line 12-13.

Lines 26 and 432: change “drived” to “drove”.

Response: Thank you for identifying this grammatical error. We have corrected "drived" to "drove" in the revised abstract in Line 27 and 488.

Lines 53–54: Ensure that all abbreviations are defined when first introduced.

Response: We sincerely appreciate your meticulous attention to abbreviation formatting. In the revised manuscript, we have systematically ensured that all abbreviations are explicitly defined upon their first mention. Key examples include: Net Ecosystem Productivity (NEP) and Carnegie-Ames-Stanford Approach (CASA) in the revised abstract in Line 11-13, Net Primary Productivity (NPP)(in Line 45 ), soil heterotrophic respiration (Rh)(in Line 46), Moderate Resolution Imaging Spectroradiometer Normalized Difference Vegetation Index (MODIS NDVI), Landsat Thematic Mapper/Operational Land Imager (TM/OLI)) Boreal Ecosystem Productivity Simulator (BEPS) in the revised introduction in Line 55-58.

Line 56: What does "Rh" refer to? Consider rewording for clarity.

Response: Thank you for highlighting this ambiguity. "Rh" refers to soil heterotrophic respiration, which represents the carbon released through the decomposition of organic matter by soil microorganisms and fauna. To enhance clarity, we have revised the text to explicitly define the abbreviation upon its first introduction in Line 46 and methods in Line 223.

Line 60: The sentence should read: ", such as rapidly urbanizing regions, remain…"

Response: Thank you for your precise suggestion to improve sentence clarity. We have revised the text as follows to incorporate the recommended phrasing in Line 64: However, the mechanisms underlying spatiotemporal differentiation of NEP in complex ecosystems, such as rapidly urbanizing regions remain underexplored.

Line 63: "Model parameterization" would be a more appropriate term.

Response: Thank you for your precise suggestion to enhance technical terminology. We have revised the text to adopt the recommended termof “model parameterization” in Line 67.

Lines 96–106: Consider moving information about Zhejiang to the "Study Area" section.

Response: Thank you for your precise suggestion. The inclusion of Zhejiang Province's profile in this context primarily aims to highlight the complexity of its ecosystems and the intensified anthropogenic pressures on vegetation carbon sinks, thereby demonstrating the critical importance of conducting this research for regional ecological security and the realization of carbon neutrality goals in economically vibrant yet ecologically vulnerable coastal regions. Thus, we choose to retain this information in this part.

Lines 131–142: Please add references for the datasets used.

Response: Thank you for emphasizing the importance of dataset citations. We have added full references for datasets in the revised data sources in Line 153-168 and references in Line 601-609.

Line 156: correct “resampling” to “resample”.

Response: Thank you for identifying this grammatical error. We have corrected "resampling" to "resample" in Line 183.

Table 1: A column for references should be included.

Response: Thank you for your precise suggestion. For datasets acquired directly from official sources and preprocessed without associated publications, citations are provided only in the main text where relevant, and these are not separately listed in tables 1.

Line 171: "Is necessary" should be used instead of "are necessary."

Response: Thank you for identifying this grammatical error. We have corrected "Is necessary" to "are necessary" in Line 204.

Lines 117-120: have excessive commas. Restructure the sentences for better readability.

Response: Thank you for your valuable suggestion to improve sentence clarity. We have restructured the text to reduce comma usage and enhance readability. The revised version in Line 137-139 now reads:The subtropical monsoon climate endows the province with mean annual precipitation ranging 1,000-2,000 mm and mean annual temperature ranging 15-18 ℃.

Line 182: Since "NPP" and "Rh" were defined in line 44, please use the abbreviations here instead of repeating the full terms.

Response: Thank you for your valuable suggestion. We have revised the text to replace the full terms with their defined abbreviations in Line 215-216 and also check other place to ensure the use of defined abbreviations.

Line 201: The alignment of the Slope formula is inconsistent. Ensure all mathematical formulas are uniformly aligned and clearly formatted.

Response: Thank you for your valuable suggestion. We have rigorously standardized the alignment and presentation of all equations throughout the manuscript in Line 233-236.

Add proper spaces between numerical data and units (e.g., change “30m” to “30 m”, “4km” to “4 km”, “300m” to “300 m”).

Response: Thank you for your valuable suggestion. We have checked and added proper spaces between numerical data and units in full text.

For consistency, add “vegetation” before all NEP mentions throughout the text, og rename the abbreviation.

Response: Thank you for your valuable suggestion. Combined with the suggestion (Line 48: In this sentence “mapping regional vegetation NEP dynamics”, I think it is better to use the word territory or ecosystem instead of vegetation. Because NEP involves all the ecosystem processes. The same in line 51 and other lines.) of the reviewer 1, NEP reflects whether the entire ecosystem (vegetation + soil + microbes, etc.) is a carbon sink (positive) or carbon source (negative), calculated as the difference between net primary productivity (NPP) and soil heterotrophic respiration (Râ‚•). Since the E of NEP represents ecosystem, we will no longer add the word ecosystem or territory in front of NEP, and the full text will be changed to “NEP” instead of “vegetation NEP” to avoid ambiguity.

Comments on the Quality of English Language

I suggest the authors carefully review the manuscript for grammatical accuracy, particularly verb tense. For example, use the present tense when stating general facts. Please ensure all abbreviations are introduced in full upon first mention and used consistently throughout the manuscript.

Response:  Thanks for your insightful comments. 1) We have carefully reviewed the entire manuscript to correct grammatical inaccuracies and ensure consistent use of verb tenses. Specifically, all general facts, established theories are now presented in the present tense, and statements describing results are now presented in the past tense. 2) In the revised manuscript, we have systematically ensured that all abbreviations are explicitly defined upon their first mention. Key examples include: Net Ecosystem Productivity (NEP) and Carnegie-Ames-Stanford Approach (CASA) in the revised abstract in Line 11-13, Net Primary Productivity (NPP)(in Line 45 ), soil heterotrophic respiration (Rh)(in Line 46), Moderate Resolution Imaging Spectroradiometer Normalized Difference Vegetation Index (MODIS NDVI), Landsat Thematic Mapper/Operational Land Imager (TM/OLI)) Boreal Ecosystem Productivity Simulator (BEPS) in the revised introduction in Line 55-58.